# SpENCNN: Orchestrating Encoding and Sparsity for Fast Homomorphically Encrypted Neural Network Inference

## Abstract

Homomorphic Encryption (HE) is a promising technology for protecting user's data privacy for Machine Learning as a Service (MLaaS) on public clouds. However, the computation overheads associated with the HE operations, which can be orders of magnitude slower than their counterparts for plaintexts, can lead to extremely high latency in neural network inference, seriously hindering its application in practice. While extensive neural network optimization techniques have been proposed, such as sparsification and pruning for plaintext domain, they cannot address this problem effectively. In this paper, we propose an HE-based CNN inference framework, i.e., SpENCNN, that can effectively exploit the single-instruction-multiple-data (SIMD) feature of the HE scheme to improve the CNN inference latency. In particular, we first develop a HE-group convolution technique that can partition channels among different groups based on the data size and ciphertext size, and then encode them into the same ciphertext in an interleaved manner, so as to dramatically reduce the bottlenecked operations in HE convolution. We further develop a sub-block weight pruning technique that can reduce more costly HE-operations for CNN convolutions. Our experiment results show that the SpENCNN-optimized CNN models can achieve overall speedups of 8.37x, 12.11x, and 19.26x for LeNet, VGG-5, and HEFNet, respectively, with negligible accuracy loss.

## 1 Introduction

For the past decade, we have witnessed the tremendous progress of the machine-learning technology and the great success achieved in practical applications. Convolution Neural Network (CNN) models, for example, have been widely used for many cognitive tasks such as face recognition, medical imaging, and human action recognition. Meanwhile, there is a growing interest to deploy machine learning models on the cloud as a service (MLaaS). While cloud computing has been well recognized as an attractive solution, especially for computation intensive applications such as the MLaaS, outsourcing sensitive data and data processing on cloud can pose a severe threat to user's privacy.

Homomorphic Encryption (HE) is a promising technology for protecting user's privacy when deploying MLaaS on cloud. HE allows computations be performed on encrypted inputs and the decrypted output matches the corresponding results computed from the original inputs. Thus, a client can encrypt the sensitive data locally and send the encrypted ciphertexts to the cloud. All intermediate results will maintain encrypted, and the encrypted results sent from cloud can be correctly decrypted using the secret key hold by the client. Whlie HE can help to maintain the confidentiality for computation process on cloud effectively, one major problem has to deal with is the excessive computational cost associated with the operations over the encrypted data: HE operations (e.g. HE multiplication, additions on encrypted data) can be several (i.e., three to seven) orders of magnitude slower than the corresponding operations on plaintexts. The tremendous computational cost of HE has been the largest bottleneck that hinders its applications on cloud.

One of the most effective approaches (e.g. (Gilad-Bachrach et al., 2016; Brutzkus et al., 2019; Dathathri et al., 2019; Kim et al., 2022)) to reduce the HE computational cost is to take advantage of the single-instruction-multiple-data (SIMD) capability, supported by HE schemes, e.g. CKKS and BFV. Smart & Vercauteren (2010) initially proposed to pack multiple data elements in the plaintext

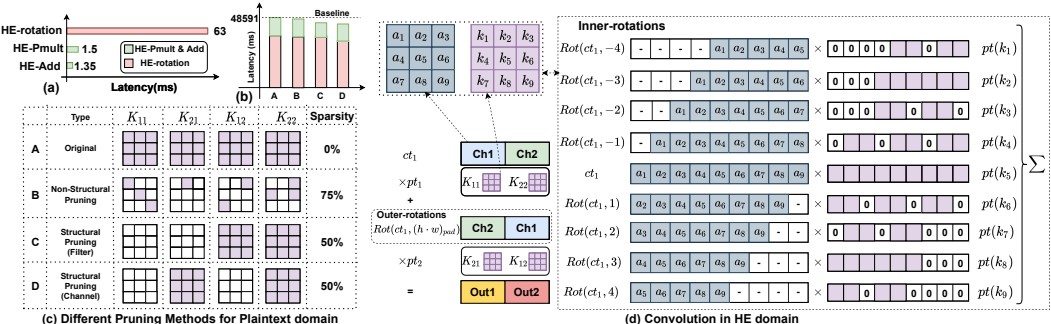

Figure 1: (a) Comparison of different HE-operations' latency (b) Comparison of the HE convolution latency under different pruning methods. (c) Illustration for different pruning methods for plaintext domain. (d) Multi-channel convolution process in HE domain. Notation definitions refer to section 3.1. $pt(k_i)$ indicate the weight plaintext. The convolution layer used here has 64 input- and 64 output-channel, with a $3 \times 3$ kernel. The input feature map size of the convolution layer is $32 \times 32$.

domain to different "slots" in the same ciphertext and thus computations for data elements at the same slot of two encoded messages can be performed in parallel. The challenge is how to pack data based on the characteristics of the applications so that computation can be conducted effectively in a SIMD scheme manner. In particular, the problem rises when the computation needs to be performed on data elements at different slots of the messages. To re-arrange the location of each individual data element in an encrypted message is out of question due to its large overhead. A more reasonable solution is to employ the HE-rotation[1] operation that can move the data element cyclically in the same message. However, HE-rotation has a high latency cost due to required permutation and key-switching operation, compared with other HE-operations such as the HE multiplication of a ciphertext with a plaintext (HE-PMult) and HE addition of two ciphertexts (HE-Add), as shown in Figure 1 (a). Therefore, how to judiciously encode the inputs and perform the SIMD operations plays the key role in reducing the HE computation complexity.

In this paper, we study the problem on how to improve the HE-based inference latency when deploying a privacy-preserving machine learning (PPML) platform based on CNN models on cloud. It is well-known that the major computation workload for CNN inference comes from the convolution layers. Assuming the user inputs (e.g. images) are encrypted as the ciphertexts and associated CNN models are encoded as plaintext messages, the major HE computations are therefore HE-PMult, HE-Add, and HE-rotation operations. Traditional neural network optimization techniques such as sparsification and pruning (Han et al., 2015b; Wen et al., 2016) help to reduce the computation demand for CNN inference for plaintext domain. However, they may not be effective here as reducing the computation demand does not necessarily imply the reduction of SIMD computations. In addition, note that, as shown in Figure 1(a), the computation cost for an HE-rotation can be over $43\times$ of that for an HE-Pmult or an HE-Add operation. Simply reducing the HE operations without optimizing the HE-rotations may not be effective at all in reducing the computational cost.

To this end, we develop an HE-based CNN inference framework, i.e., SpENCNN, with the goal to effectively exploit the SIMD feature of the HE scheme to improve the CNN inference latency. In particular, we develop two techniques to reduce the HE computational cost. First, we develop HE-group convolution and associated group-interleaved encoding to optimize channel locations on ciphertexts based on the number of convolutional groups and ciphertext size, thus significantly reducing the number of costly HE-rotations. Second, we further optimize the model architecture by pruning and training the weights in the sub-blocks iteratively with the goal to minimize HE-rotations and accuracy loss. We have conducted extensive experiments based on three CNN models on MNIST dataset and CIFAR-10 dataset and results show that the optimized CNN models can achieve overall speedups of 8.37x, 12.11x, and 19.26x for LeNet, VGG-5, and HEFNet, respectively, with negligible accuracy loss. To our best knowledge, this is the first work to that builds optimizing framework for CNN model architecture from the aspect of structural sparsity and data packing in HE to benefit HE-based PPML inference.

---

[1]For instance, $Rot(ct, k)$ transforms an encryption of $(v_0, ..., v_{N/2-1})$ into an encryption of $(v_k, ..., v_{N/2-1}, v_0, ..., v_{k-1})$

## 2 PRELIMINARIES

### 2.1 CKKS HOMOMORPHIC ENCRYPTION

Homomorphic Encryption (HE) allows computations to be performed on encrypted data without decryption. Among various HE schemes, the levelled HE– Cheon-Kim-Kim-Song (CKKS) (Cheon et al., 2017) is widely adopted in the encrypted neural network inference because of supporting the fixed-point real number arithmetic and potentially avoiding the prohibitively expensive bootstrapping. The CKKS-based HE operations mainly consist of ciphertext addition HE-Add $(ct_1 + ct_2)$, ciphertext multiplication HE-Cmult *(ct₁ × ct₂)*, scalar multiplication HE-Pmult *(pt₁ × ct₂)*, ciphertext roation HE-rotation *Rot(ct, k)*, etc. For MLaaS that only encrypts clients' data, HE-Add, HE-Pmult and HE-rotation often dominate the computations of an encrypted inference. Among these three operations, HE-rotation costs much longer latency than the other two, e.g. $\sim 43\times$ as our profiling result in Figure 1 (a) shows, due to the complex automorphism operation and a key-switching operation. The detailed calculation process of HE-rotation can be described as:

$$Rot(ct, k) = (c(X^{ik}), 0) + P^{-1}(a(X^{ik}) \cdot evk_{rot}^k) \tag{1}$$

where the evaluation key $(evk_{rot}^k)$ is a public key with a larger modulus $PQ$, and $P$ is greater than $Q$. Assume $ct = (c(X^i), a(X^i))$ represents a ciphertext before rotation, then the automorphism $(c(X^{ik})$ and $a(X^{ik}))$ maps each polynomial coefficient index $i$ to output polynomial coefficient index $ik\ mod\ N$, where $N$ is the polynomial degree. The second term on the right side of Equation 1 represents the key-switching operation to ensure the final ciphertext can be still decrypted by the same secret key. It is very expensive and could take over 90% of all operations in practice (Samardzic et al., 2022)

### 2.2 THREAT MODEL

We assume the cloud-based machine learning service, of which a trained convolutional neural network (CNN) model with plaintext weights, is hosted in a cloud server. A client could upload his/her private and sensitive data to the public cloud for obtaining an online inference service. The cloud server is semi-honest (e.g. honest but curious). To ensue the confidentiality of clients' data against such a cloud server, the client utilizes HE to encrypt the data and then send it to cloud for performing encrypted inference without decrypting the data or accessing the private key. Finally the client can decrypt the returned encrypted inference results from cloud using a private key. In this work, we focus on encrypting the client's data and others like model parameters, are assumed as plaintext.

### 2.3 MOTIVATION EXAMPLE

To identify the computation bottleneck in HE inference, we analyze the computation pattern of the convolutional layer, which often dominates CNN inference's memory and computational overheads, in the encryption process. Here the input and output activation feature maps are encrypted as ciphertext, while the convolutional kernels are assumed as plaintext. We also assume the state-of-the-art ciphertext encoding–row-major (Dathathri et al., 2019; Kim et al., 2022) is adopted here. This allows efficient multi-channel ciphertext packing to take advantage of CPU's single-instruction-multiple-data (SIMD) architecture for fast HE inference. Figure 1 (d) shows the typical HE convolution process of a convolution layer which consists of 2-input/output channels with $3 \times 3$ kernels. To compute a ciphertext output feature map, two types of cipertext rotations need to be performed sequentially. First, **inner-rotation** rotates each input channel's ciphertext feature map 8 times (or $K^2 - 1$, here kernel size $K = 3$). Each rotated version will need to be multiplied with its corresponding weight plaintext, and then such results will be summed up to obtain an intermediate ciphertext from each input channel, which will further be concatenated as a whole ciphertext (e.g. Ch1 and Ch2 as **ct1**). Second, **outer-rotation** rotates the concatenated ciphertext multiple times (in this simple example, 1 time because of packing 2 output channels as a ciphertext). Finally all ciphertext output feature maps can be obtained in parallel by the summation of these rotated copies. Apparently, compared to non-encrypted convolution, HE convolution significantly escalates the memory and computation overheads. Moreover, since the latency of HE-rotation can be much higher than other operations due to complex automorphism and key switching operations (see our profiling result in Figure 1(a) 63ms for HE-Rot v.s. 1.5ms for HE-Pmult, detailed setting in Sec. 4.1), and the multi-channel convolutions in deep CNNs would involve a huge volume of HE-rotation [2]. As a result, the long-latency HE-rotation quickly becomes a bottleneck of the encrypted inference.

---

[2]The matrix-vector multiplications in fully-connected layers also require a substantial amount of HE-rotation.

Figure 2: The overall flow of SpENCNN framework for optimizing HE-based CNN inference, mainly consists of two orthogonal techniques to generate a tailored CNN model: (1) HE-group Convolution (outer-rotation optimization), and (2) the Sub-block Pruning (inner-rotation optimization).

One straightforward solution to accelerating HE inference is to reduce the number of rotations through zeroing out (or pruning) the plaintext weights. As Figure 1(c) shows, if any weight plaintext–$pt(k_i)$ contains all zero values, then the corresponding ciphertext rotation–$Rot(ct_1, k)$ and its associated multiplication and summation can be safely eliminated. Since existing pruning techniques have been proved to be effective in reducing the computation and memory overhead to speedup the nonencrypted inference without accuracy drop, we apply two representative pruning methods–non-structured pruning(Han et al., 2015a) (zeros appear randomly in a kernel, see Fig. 1 (c)–B, 75% sparsity) and structured pruning(Wen et al., 2016) (structured zeros in a kernel, see filter pruning and channel pruning in Fig. 1 (c)–C and D, 50% sparsity). For HE operations in an example convolutional layer with 64 input/output channels, feature map size $32 \times 32$ and kernel size $3 \times 3$, as Fig. 1 (d) shows, the existing pruning achieves very marginal or even no reduction of the HE-rotation latency which dominates the convolution computation. In the worst case, it even cannot remove any HE-rotation despite the high model sparsity, e.g. the non-structured pruning with 75% sparsity ratio. *The underlying reason is two-fold: 1) pruning is unable to address the outer-rotation since computing an output ciphertext by convolution needs to sum all channels' feature maps belonging to the same ciphertext if using the state-of-the-art ciphertext encoding (see Fig. 3 (a)); 2) existing pruning techniques are designed for non-encryted inference, and the special channel-wise ciphertext operations involved in HE convolution are ignored.* This prompts the need of jointly optimizing ciphertext encoding and encryption-aware model sparsity to accelerate HE inference.

## 3 THE SPENCNN FRAMEWORK

In this section, we present the technical details of our proposed SpENCNN framework. Figure 2 depicts its overall flow. The SpENCNN framework takes an initial CNN model as input and outputs a tailored CNN model after two processing stages–**(1) HE-group Convolution** is designed to reduce the outer-rotations caused by multi-channel convolution. In particular, we design an adjustable method and determine a theoretically optimal group number $G_{base}$ based on the size of the ciphertext and data packed in the CKKS HE scheme, to ensure that all outer-rotations can be eliminated while keeping model accuracy. **(2) Sub-block Pruning** is further proposed to reduce the number of inner-rotations. However, this is not trivial. We observe that to reduce as many inner-rotations as possible, we must precisely identify and completely prune selected sub-blocks. This, unfortunately, would result in an considerable accuracy drop. To address this issue, we develop a set of sub-steps which include identifying subblocks, pruning the subblocks, updating the remaining subblocks, and retraining the model for accuracy recovery.

### 3.1 HE-GROUP CONVOLUTION

**Two intuitions.** Our proposed HE-group convolution is based on two intuitions. We observe that the number of required outer-rotations for a ciphertext is $Rot_{outer} = {}^{N}/_{2 \times (h \times w)_{pad}} - 1$, where $N$ is the polynomial degree defined in the cryptographic parameters, $h$ and $w$ are the height and width of the input feature map, and $pad$ rounds a number to the next power of two. Each ciphertext generated by the outer-rotation further requires a set of inner-rotations. The number of inner-rotations is $Rot_{inner} = K^2 - 1$, where $k$ is the convolutional kernel size. Apparently, increasing the number of outer-rotation–$Rot_{outer}$ by just 1 can bring an extra $Rot_{inner} = K^2 - 1$ inner-rotations. This gives us the first intuition–*reducing the number of outer-rotations will fundamentally reduce the computational overhead.*

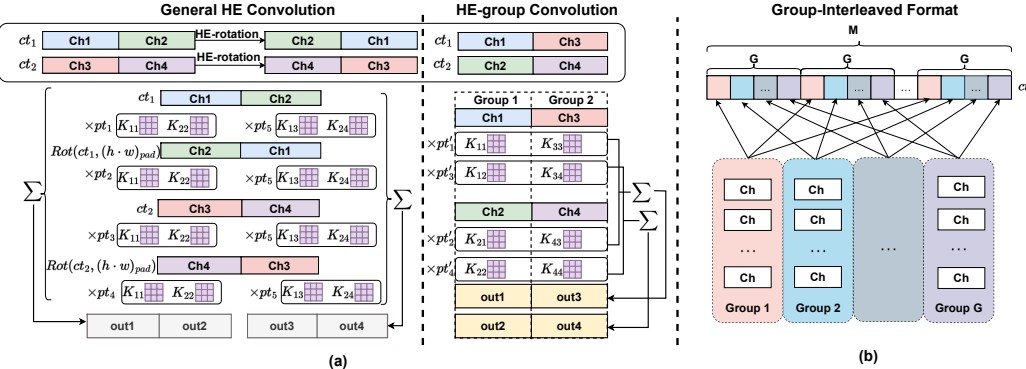

Figure 3: (a) an example of HE convolution which generates 4 output channels from the 4 input channels with $3 \times 3$ kernels. By HE-group convolution, we only need 2 group-interleaved encoded cts multiply with corresponding pts and sum up to get the output channels data. (b) proposed HE-group convolution by a group-interleaved format.

The reason behind the outer-rotation is that multiple channels on the same ciphertext are involved in the same multichannel convolution. In our study, we find that convolution (e.g., depthwise convolution (Howard et al., 2017)) can be also performed individually within each single channel. We also find that the group convolution technique (Krizhevsky et al., 2012; Zhang et al., 2018; Ioannou et al., 2017) can reduce the number of channels involved in each group. This gives us the second intuition–*reducing the number of channels in the same group can eliminate the outer-rotation*.

Based on these intuitions, we design the HE-group convolution. Given the group number $G$, we have the upper bound of the number of channels in the same group as $\lceil (N/2 \cdot (h \cdot w)_{pad}) \times 1/G \rceil$. Then the relationship between $Rot_{outer}$ and $G$ can be expressed as:

$$Rot_{outer} = \lceil (N/2 \cdot (h \cdot w)_{pad}) \times 1/G \rceil - 1 \qquad (2)$$

Accordingly, we can use an appropriate $G$ value to cancel the first term, i.e., the $G_{base} = N/2 \cdot (h \cdot w)_{pad}$ in our design. Theoretically, this optimized value indicates zero outer-rotations.

**Group-interleaved encoding.** However, we find that the traditional ciphertext encoding format is not compatible when we implement our grouping idea. This is because the row-major format is mainly designed to perform convolution in the SIMD manner without considering the channel positions on the ciphertext. To address this issue, we propose a **group-interleaved encoding** format–the channel data from different groups are placed on the same ciphertext in an interleaved manner. This new encoding facilitates fast HE-group convolution without involving any outer-rotation.

Figure 3 shows an example of proposed HE-group convolution and group-interleaved encoding. We assume that two ciphertexts contain 4 channels of data, and each ciphertext $ct_i$ contains 2 channels. As shown in Figure 3 (a)–left, in general HE convolution, each $ct_i$ has to do 1 outer-rotation to cover the 2 different channels, i.e., $\{ch1, ch2\}$ and $\{ch2, ch1\}$ for $ct_1$. Convolution will be performed individually on each outer-rotated case for all ciphertexts. The encrypted output channels $\{out_1, out_2\}$, $\{out_3, out_4\}$ can be generated after a summation.

For our HE group convolution, as Figure 3 (a)–right shows, sibling channels $\{ch1, ch2\}$ and $\{ch3, ch4\}$ from the same $ct_i$ are in different convolution groups, which are encoded by our group-interleaved format, i.e., $\{ch1, ch3\}$ in $ct_1$ and $\{ch2, ch4\}$ in $ct_2$. Now, the outer rotation is eliminated because each $ct_i$ can perform 2 groups of convolution individually without rotating the channels. The encrypted output channels after summation, i.e., $\{out_1, out_3\}$, $\{out_2, out_4\}$, are naturally group-interleaved and can immediately send to the next HE-group convolution. Figure 3 (b) further shows the generalized group-interleaved encoding, in which a ciphertext can encrypt $M$ channels using the adjustable convolution group number $G$, with constraints $G \le M$ and $M\%G = 0$.

## 3.2 SUB-BLOCK PRUNING

We design the sub-block pruning to further remove the remaining inner-rotations after the HE-group convolution. Our idea is to prune (zero out) a whole set of weights corresponding to specific inner-rotations, so that the computational overhead of these inner-rotations can be eliminated. This reminds us of the weight sparsity in CNN models. In HE-convolution, an inner-rotated ciphertext will be

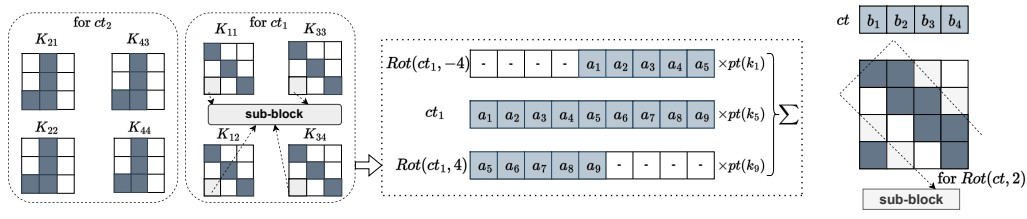

(a) Weight sparsity in convolutional layers    (b) Weight sparsity in FC layers

Figure 4: (a) The weight sparse pattern in convolutional layers. For the same $ct$, their weight sparse patterns must be same. For different $ct$, the weight sparse pattern may change. (b) The weight sparse pattern in FC layers is in a diagonal-wise shape.

multiplied with the weights at the same position, namely "sub-block", from all relevant kernels. Therefore, our design tends to cut out the same sparse pattern on these sub-blocks for all relevant kernels of the same ciphertext.

As the example in Figure 4 (a) shows, the 4 kernels of $ct_1$ shares the same sparse pattern, thus eliminating 6 inner-rotations. This pruning scheme can be also extended to the FC layer. As shown in Figure 4 (b), the multiplication of ciphertext on FC layer is equivalent to the convolution using a diagonal-wise encoding method (Halevi & Shoup, 2014). Multiplying with the weights in one diagonal line requires one HE-rotated copy of ciphertext. (see Appendix A.3 for more details.)

To obtain the desired sparse pattern, we propose the sub-block pruning. The more sub-blocks $p_i$ being pruned, the less inner-rotations needed. This is actually an optimization problem, in which, we need to minimize the total number of sub-blocks while maintaining the prediction accuracy concurrently:

$$\min\{P = \sum_i p_i \cdot I_i\} \text{ s.t. } Acc(f(x;(W,P))) \geq Acc(f(x;W)) \text{ where } I_i = \begin{cases} 0 & p_i \text{ is pruned} \\ 1 & other \end{cases} \quad (3)$$

We would like to find the sub-block with the minimum weight importance to the model at each iteration and prune, then, retrain the model for a few epochs to recover the accuracy. However, the sizes of sub-blocks are different. To measure the weight importance of each sub-block in a fair way. we define the weight importance metric as an average $L^2$ norm $\frac{||w_{p_i}||}{dim(w_{p_i})}$. As described in algorithm 1, at each iteration, we would like to prune the sub-block with the least weight importance. The iterative algorithm would stop when the model accuracy is lower than the initial accuracy.

## 4 EVALUATION

### 4.1 EXPERIMENT SETUP

**Setup.** We conduct our experiments on a workstation equipped with an AMD Ryzen Threadripper 3975WX CPU, an NVIDIA RTX 3090 GPU, and 256GB of RAM. To evaluate our proposed SpENCNN, we select three baseline CNN models that are often adopted in HE inference performance evaluation, and implement them using PyTorch on GPU. This includes LeNet-like for MNIST

| Model | # Layers | | | Groups | Accuracy |
| --- | --- | --- | --- | --- | --- |
| | Conv | FC | Act | $(G_{base})$ | (%) |
| LeNet-like | 2 | 2 | 3 | 4 | 98.95 |
| VGG-5 | 3 | 3 | 5 | 8 | 84.06 |
| HEFNet | 4 | 1 | 4 | 8 | 83.67 |
| Model | Encryption Parameters | | | Mult | Security |
| | N | P | Q | Level | Level |
| LeNet-like | 8192 | 264 | 24 | 10 | >128 bit |
| VGG-5 | 16384 | 529 | 31 | 16 | > 80 bit |
| HEFNet | 16384 | 436 | 31 | 13 | >128 bit |

Table 1: Three baseline Convolutional Networks models and corresponding Encryption Parameters. LeNet-like is for the MNIST dataset. VGG-5 and HEFNet are for CIFAR-10 dataset.

---

**Algorithm 1** Sub-blocks Iterative Pruning

1: **Input: CNN model:** $f(x;(W,P))$,
2: **Remark:** x-Data, W-Weights, P-HE blocks
3: **Output: HE-friendly model:** $f(x;(W',P'))$
4: $P' = P = \sum_i p_i \cdot I_i$
5: **While Accuracy loss** $\leq 0$ :
6: $\quad i \leftarrow \underset{i}{\operatorname{argmin}} \frac{||w_{p_i}||}{dim(w_{p_i})}$
7: $\quad I_i = 0$
8: $\quad$ **prune weights in** $p_i$ **from current** $P$
9: $\quad$ **update** $P'$
10: $\quad$ **retrain** model with $P'$ and update $W'$
11: **end While**
12: **Return** $f(x;(W',P'))$

dataset (2016), VGG-5 (Rathi et al., 2020) and the HE-friendly Net (HEFNet) for CIFAR10 dataset (Details of layer size is in Appendix A.2). Since the non-linear activation function like ReLU cannot be evaluated in HE, we replace ReLU with the adaptive quadratic polynomial function $f(x) = ax^2 + bx + c$ following the related works (Dathathri et al., 2019; Kim et al., 2022), where $a, b, c$ are trainable parameters to maintain the model accuracy. Table 1 lists the specifications of these three models and the corresponding accuracy. In particular, the test accuracy for LeNet-like-MINST, VGG-5-CIFAR10, HEFNET-CIFAR10, is 98.95%, 84.06%. and 83.67%, respectively, which are consistent with their original versions.

We use Microsoft SEAL library v3.4.5 (SEAL) to implement the RNS-CKKS HE computation on these networks. Table 1 also lists the key parameters used in our RNS-CKKS encryption, including the polynomial degree $N$, the total modulus in bit-length $Q$, the scale factor in bit-length $P$ to maintain the HE evaluation accuracy, and total multiplication level. These parameters can guarantee a security level of 80 bit for VGG-5, 128 bit for LeNet-like and HEFNet.

**Methodology.** We first perform an ablation study to evaluate each individual technique's effectiveness, and then compare the whole SpENCNN framework with the state-of-the-art method. y We adopt the average inference latency (in seconds) as the main measurement. An image set containing 20 different samples is used to measure and report the latency for these models. A lower latency indicates better performance. In addition, we measure the left holomorphic operation count (HOC, in %), sparsity (in %), and accuracy (in %) on the tailored models. The lower HOC and lower sparsity while offering higher accuracy are desired on all models.

## 4.2 RESULTS

### 4.2.1 EVALUATON ON HE-GROUP CONVOLUTION

Table 2 lists our evaluation results for the HE-group convolution. We apply the HE group convolution alone (in ablation) to each baseline model and evaluate its effectiveness and scalability. In particular, we adjust the number of groups from its default (i.e., 1-baseline) until it exceed its $G_{base}$ according to our design (i.e., the highlighted 4, 8, and 8 for LeNet-like, VGG-5, and HEFNet, respectively). For detailed analysis, we breakdown HOC into "Rot" (HE-rotation) and "Others" (other operations including HE-Pmult and HE-add).

Our HE-group convolution can be scaled to any convolutional model. As the number of groups increases, it can effectively reduce the number of HOC and maintain the accuracy, thus reducing the latency of HE inference and improving the performance. As listed in Table 2, the number of HE-rotation is reduced from 100% to 27.27%, 85.45%, and 11.95% on LeNet-like, VGG-5, and HEFNet, respectively. Once the $G_{base}$ is reached, the number of HE-rotation does not decrease further in spite of increasing the number of groups. This is because the outer-rotation is completely eliminated in HE-group convolution after the group number reaches $G_{base}$. We also find that HE group convolution can reduce other HOC such as HE-Pmult and HE-add even after exceeding $G_{base}$. This also contributes to the performance improvement.

For example, the HE-group convolution is particularly effective on our HEFNet (i.e., $\sim 88\%$ and $\sim 86\%$ reduction for HE-rotation and others, respectively). This is because it has the largest volume of convolution layers among the three models. Such a dramatic reduction in HOC further shortens the

| Model | Groups | HOC Left (%) | | Accuracy (%) | Latency (s) | Speedup (×) |
| --- | --- | --- | --- | --- | --- | --- |
| | | Rot | Others | | | |
| LeNet-like | 1-baseline | - | - | 98.95 | 1.2658 | - |
| | 2 | 51.52 | 52.91 | 98.95 | 0.6806 | 1.86 |
| | **4** | **27.27** | **28.24** | **98.95** | **0.3807** | **3.32** |
| | 8 | 27.27 | 16.47 | 98.67 | 0.3044 | 4.16 |
| VGG-5 | 1-baseline | - | - | 85.16 | 53.909 | - |
| | 4 | 87.53 | 84.08 | 84.53 | 46.539 | 1.16 |
| | **8** | **85.45** | **81.42** | **84.06** | **45.311** | **1.19** |
| | 16 | 85.45 | 80.10 | 82.23 | 45.053 | 1.20 |
| HEFNet | 1-baseline | - | - | 84.91 | 24.113 | - |
| | 4 | 24.53 | 25.74 | 84.35 | 6.2491 | 3.86 |
| | **8** | **11.95** | **13.36** | **83.67** | **3.2718** | **7.37** |
| | 16 | 11.95 | 7.18 | 80.06 | 2.3627 | 10.21 |

Table 2: Ablation study of HE-group convolution with different number of convolution groups.

inference latency from 24.11s to 3.27s, which represents a $7.37\times$ speedup. In contrast, the HE group convolution is the least effective in VGG-5, as it contains three large FC layers (size of $8192\times4096$), which cannot be substantially optimized using HE group convolution alone. There are more than 85% of HE-rotations and 80% of other operations that cannot be eliminated. And this number saturates as a lower bound after reaching the $G_{base}$, resulting in a limited speedup of $1.19\times$.

We also observe that the model accuracy slightly decreases as the number of groups increases. This is due to the fact that fewer channels are involved in the HE-group convolution compared to the general convolution (see Figure 3). Fortunately, as long as the number of groups does not exceed our suggested $G_{base}$, the loss of accuracy is marginal (i.e., 0%, 1.1%, and 1.2% on LeNet-like, VGG-5, and HEFNet, respectively). Also, our design is adjustable, allowing a trade-off between the accuracy and the optimized number of convolution groups.

### 4.2.2 EVALUATION ON SUB-BLOCK PRUNING

Here, we apply the sub-block pruning alone (in ablation) and compare it with other pruning methods such as Non-structural prune (Han et al., 2015a), and Structural-prune (Wen et al., 2016). Table 3 lists our evaluation results. We do not include accuracy in this evaluation since we prune each baseline model using each pruning method under the constraint of maintaining the original model accuracy. Instead, we include the sparsity (i.e., the percentage of pruned weights) for comparison.

Our sub-block pruning can effectively improve the HE inference performance on all baseline models (i.e., the speedup of $2.62\times$, $6.15\times$, and $2.57\times$ on LeNet-like, VGG-5, and HEFNet, respectively), which significantly outperforms other traditional pruning methods (i.e., marginal $\sim 1.1\times$ speedup on most cases). The reason is obvious but significant. Our design is more HE-oriented and effective in the ciphertext domain while traditional pruning is more sparsity-oriented and for the plaintext computation efficiency only. For example, although NS-prune can prune $\sim 92\%$ of the weights on VGG-5, it cannot eliminate the HE overhead (i.e., $\sim 96\%$ HOC) caused by the remaining $\sim 8\%$ of the weights.

Our sub-block pruning method performs the best (i.e., $\sim 16\%$ HOC) on VGG-5 because it effectively eliminates the inner-rotations caused by the large number of redundant weights in the FC layers. Together with the previous results (see Table 2), sub-block pruning can be a good complement to the HE-group convolution that performs weakly on the FC layers. We also note that our sub-block pruning method on LeNet-like (i.e., 35.21% Rot left) slightly outperforms HEFNet (i.e., 41.88% Rot left). This is because the larger convolutional kernel (i.e., $5 \times 5$) in LeNet-like gives our pruning method more space to optimize the inner-rotations.

### 4.2.3 COMPARE WITH THE STATE-OF-THE-ART

We compare our method with the state-of-the-art HE-prune method–Hunter (Cai et al., 2022). The comparison results are presented in Table 4. In this evaluation, our method combines the HE-group convolution and the sub-block pruning, and uses the proposed $G_{base}$ as the group number (i.e., highlighted data). For a fair comparison, pruning is well controlled to ensure that our method and Hunter have the same level of accuracy (i.e., error$\leq \pm 0.04\%$) on all baseline models. We can see from Table 4, our method outperforms the state-of-the-art significantly in terms of HOC, sparsity, and

| Network | Groups | HOC Left (%) | | Sparsity (%) | Latency (s) | Speedup ($\times$) |
| | | Rot | Others | | | |
| --- | --- | --- | --- | --- | --- | --- |
| LeNet-like | Dense-Baseline | - | - | 0.00 | 1.2658 | - |
| | NS-prune | 96.12 | 96.23 | 91.00 | 1.2190 | 1.04 |
| | S-prune (channel) | 88.03 | 92.82 | 53.77 | 1.1202 | 1.13 |
| | **Sub-block prune** | **35.21** | **34.07** | **63.83** | **0.4644** | **2.62** |
| VGG-5 | Dense-Baseline | - | - | 0.00 | 53.909 | - |
| | NS-prune | 97.59 | 97.14 | 91.88 | 52.5280 | 1.03 |
| | S-prune (channel) | 98.47 | 98.08 | 90.48 | 50.7178 | 1.06 |
| | **Sub-block prune** | **15.89** | **16.11** | **89.87** | **8.7659** | **6.15** |
| HEFNet | Dense-Baseline | - | - | 0.00 | 24.113 | - |
| | NS-prune | 85.60 | 88.97 | 72.95 | 21.1660 | 1.14 |
| | S-prune (channel) | 94.69 | 95.24 | 51.91 | 22.9240 | 1.05 |
| | **Sub-block prune** | **41.88** | **36.11** | **63.90** | **9.3709** | **2.57** |

Table 3: Ablation study of sub-block prune and comparison with other pruning methods.

| Network | Method | HOC Left (%) | | Sparsity (%) | Accuracy (%) | Latency (s) | Speedup (×) |
|---------|--------|------|--------|--------------|--------------|-------------|-------------|
| | | Rot | Others | | | | |
| LeNet-like | Baseline | - | - | 0 | 98.95 | 1.2658 | - |
| | Hunter | 40.95 | 39.91 | 59.99 | 98.95 | 0.5353 | 2.36 |
| | **Ours-4** | **8.54** | **9.88** | **62.62** | **98.95** | **0.1535** | **8.37** |
| VGG-5 | Baseline | - | - | 0 | 85.16 | 53.909 | - |
| | Hunter | 17.86 | 18.93 | 89.81 | 84.03 | 9.9916 | 5.40 |
| | **Ours-8** | **7.86** | **7.72** | **91.97** | **84.07** | **4.3830** | **12.11** |
| HEFNet | Baseline | - | - | 0 | 84.91 | 24.113 | - |
| | Hunter | 48.27 | 42.20 | 57.82 | 83.63 | 10.855 | 2.22 |
| | **Ours-8** | **3.99** | **4.61** | **65.62** | **83.67** | **1.2520** | **19.26** |

Table 4: Comparison with Hunter on model HOC left, sparsity, accuracy,latency ,and speedup.

latency, on all baseline models. For example, our method eliminates $96\%$ HE-rotations on HEFNet, achieving a $19.26\times$ speedup. In contrast, the Hunter-optimized model still has $51\%$ HE-rotations left behind and achieves only $2.22\times$ speedup compared to the un-pruned baseline. This is because our method is designed to eliminate both outer and inner HE-rotations by synthetically applying the HE-group convolution and sub-block pruning, while the state-of-the-art is solely built upon the fixed structure pruning. We also print out the sparse pattern in a convolutional layer of LeNet-like, included in Appendix A.1.

## 5 RELATED WORK

CryptoNets (Gilad-Bachrach et al., 2016) is an initial attempt to realize HE-inference. After that, many subsequent works are proposed to improve the HE-inference latency from different aspects. Faster-CryptoNets (Chou et al., 2018) combines weight pruning and quantization to obtain a sparse polynomial representation to speed up the PMult operation, which achieves $6.38\times$ latency reduction. LoLa (Brutzkus et al., 2019) successfully demonstrates the HE inference on a simple 3-layer model (1 convolutional layer and 2 FC layers) and achieves a 2.2s inference latency on the MNIST sample by leveraging HE schemes, data encoding format, and rotation techniques. CHET and HEAR (Dathathri et al., 2019; Kim et al., 2022) further refine the row-major coding format to achieve the same level of inference latency, but on a larger network (3 convolutional layers and minimum 64 channels).

Lou & Jiang (2021) propose a neural architecture search (NAS) based method to reduce the encryption parameters and speed up the HE-inference. Further, Ghodsi et al. (2020); Jha et al. (2021); Mishra et al. (2020); Lou et al. (2020) propose to reduce the cost of non-linear operations in NAS based HE-inference since operations like ReLU dominate the latency in the multi-party computation (MPC) setting. HE-PEx (Aharoni et al., 2022) and Hunter (Cai et al., 2022) attempt to structurally prune the weights to accelerate the HE-inference. In Hunter, a structural pruning method is proposed to facilitate HE in the MPC setting. HE-PEx adpots Hunter's method to prune the weights in the FC layer only. It can reduce memory requirement and latency by 60% on the tested autoencoder models. Our work–SpENCNN is the first to orchestrate the ciphertext encoding and model sparsity design for HE inference acceleration, significantly outperforming these works.

## 6 CONCLUSION

In this paper, we propose a fast LHE-based encrypted inference framework-SpENCNN built upon two novel techniques–HE-group convolution and sub-block weight pruning. Experimental results show that our solution can speed up the privacy-preserving inference by $8.37\times$, $12.11\times$, and $19.26\times$ on LeNet-like, VGG-5, and HEFNet, respectively, greatly outperforming the state-of-the-art solutions. In the future, we would like to extend our work to deeper models and complex classification tasks in a no-client-interaction setting by leveraging bootstrapping. We hope to apply our framework to these deep models. There shall exist potential optimizations for the trade-off between model sparsification, data encoding, bootstrapping, and cryptographic parameters.

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

# A APPENDIX

## A.1 SPARSE PATTERNS

We present the weights after optimized in binary representation. In each group, it contains 64 weight kernels with size $5 \times 5$. Kernels in same group indicate that they are associated with the same ciphertext. Within the same group, across different ciphertexts, the sparse patterns are different.

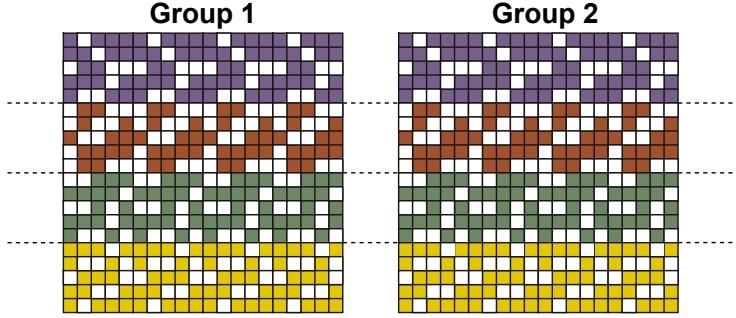

Figure 5: The sparse patterns for weight kernels in LeNet-like 2nd Convolutional layer.

## A.2 BASELINE MODEL ARCHITECTURE IN DETAIL

The following Table 5 contain the detailed convolutional kernels size, weight matrix size and number of channels. These three baseline models has different property. The LeNet-like model is a tiny model designed for simple classification task so that the channel number and weight matrix size is small, and has the least number of layers. The VGG-5 model has much more weights in FC layers and the max number of layers. The HEFNet contain the most convolutional layers and the widest channel size.

| Network | Layer | # Input channel | # Output channel | Kernel size (Matrix size in FC) |
|---|---|---|---|---|
| LeNet-like | Conv1 | 1 | 32 | 5×5 |
| | Conv2 | 32 | 64 | 5×5 |
| | FC1 | 64 | 32 | 64×32 |
| | FC2 | 32 | 10 | 32×10 |
| VGG-5 | Conv1 | 3 | 64 | 3×3 |
| | Conv2 | 64 | 128 | 3×3 |
| | Conv3 | 128 | 128 | 3×3 |
| | FC1 | 8192 | 4096 | 8192×4096 |
| | FC2 | 4096 | 4096 | 4096×4096 |
| | FC3 | 4096 | 10 | 4096×10 |
| HEFNet | Conv1 | 3 | 64 | 3×3 |
| | Conv2 | 64 | 128 | 3×3 |
| | Conv3 | 128 | 256 | 3×3 |
| | Conv4 | 256 | 256 | 3×3 |
| | FC1 | 1024 | 10 | 1024×10 |

Table 5: Convolutional layer and Fully-connected layer size in three baseline models.

## A.3 MATRIX MULTIPLICATION IN FC LAYER

Halevi & Shoup (2014) proposed a diagonal-wise multiplication of ciphertexts on FC layers. Given a N-element ciphertext $ct = \{x_{1..N}\}$, the weights of FC layer can be reshaped into a $M \times N \times N$ tensor $\{\vec{W}\}$ (will pad with zero if needed). For each $N \times N$ matrix $W$, we can multiply the diagonal $N$ elements (as plaintext) with the rotated copies of ciphertext. As the example shown in Figure 6, we have $ct = \{x_{1..4}\}$ and a $4 \times 4$ matrix $W$. We have $pt = \{a_{11}, a_{22}, a_{33}, a_{44}\}$,

$pt_1 = \{a_{41}, a_{12}, a_{23}, a_{34}\}$, $pt_2 = \{a_{31}, a_{42}, a_{13}, a_{24}\}$, and $pt_3 = \{a_{21}, a_{32}, a_{43}, a_{14}\}$. They will be multiplied by $ct = \{x_1, x_2, x_3, x_4\}$, $Rot(ct, 1) = \{x_2, x_3, x_4, x_1\}$, $Rot(ct, 2) = \{x_3, x_4, x_1, x_2\}$, and $Rot(ct, 3) = \{x_4, x_1, x_2, x_3\}$, respectively and then summed.

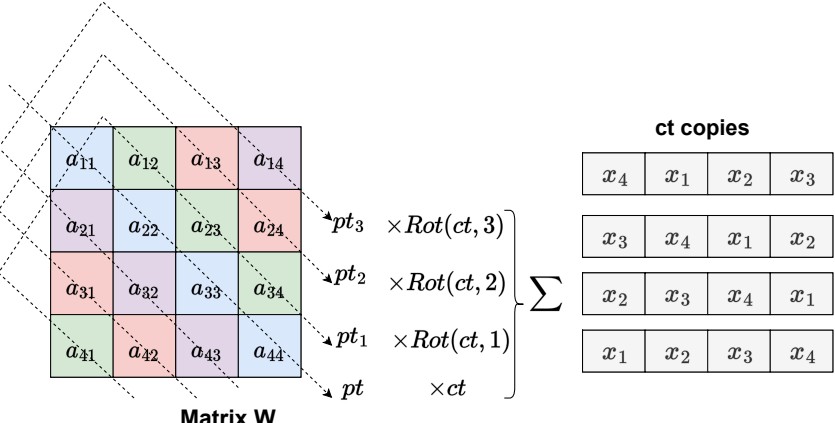

Figure 6: Ciphertext multiply with a matrix.

