# OpenReview forum: "SpENCNN: Orchestrating Encoding and Sparsity for Fast Homomorphically Encrypted Neural Network Inference"
_ICLR.cc/2023/Conference — Submitted to ICLR 2023_

### Official Review · Reviewer_WT91 · 2022-10-24

**Confidence:** 3
**Correctness:** 4
**Technical Novelty And Significance:** 2
**Empirical Novelty And Significance:** 4
**Recommendation:** 5

**Clarity, Quality, Novelty And Reproducibility:**

The result of this paper looks convincing to me, and their group-interleaved data encoding scheme is very interesting.

For clarity, I think the data encoding scheme deserves a better ( and more formal ) explanation. It's hard to comprehend from just a figure and a short paragraph.

**Strength And Weaknesses:**

Strength:
They considered the problem very common and important in building homomorphic encrypted neural network applications, especially the data encoding one. Also, their solutions are simple and very effective.

Weakness:
- In the experiment setup, they didn't explicitly mention how they deal with level-zero ciphertexts. I assume that they didn't use bootstrapping and do the re-encryption.
- Same as above, usually in applications, bootstrapping will dominate in the computational cost. Which will make this work less important than it is justified to be. However, these are two independent directions of improvement.
- Except for bootstrapping, the choice of activation function is also questionable. Beyond quadratic polynomials, other works already adopt high-precision approximation of the non-linear ReLU function for higher accuracy (e.g. https://arxiv.org/abs/2105.10879 ). Though, it's also independent of the proposed techniques. But it's better to check in experiments if they could be combined.

**Summary Of The Paper:**

This paper presents two techniques for speeding up convolutional neural network inference in homomorphic encryption settings.
Conceptually, the underlying RNS-CKKS homomorphic encryption provides the functionality to securely evaluate functions over a vector of encrypted data in a SIMD manner. In this specific setting, this paper proposes a novel data encoding scheme and a HE-specific pruning technique (sub-block pruning) that substantially reduce the number of homomorphic operations.

**Summary Of The Review:**

Overall, I think the problem they considered is very important, and the proposed solutions are simple and effective and have a very wide application in secure neural network applications. But this paper's contribution is also limited by the lack of consideration of other aspects like bootstrapping and approximation of non-linear activation function.

---

> ### Author Response · Authors · 2022-11-17
> **Response to Reviewer WT91**
>
> We would like to thank reviewer WT91 for his/her dedicated review comments for our work. In particular, we would like to thank the positive comments with regard to the novelty, timeliness, and effectiveness of our research. In what follows, we will present our response to the reviewer's comments and concerns.
>
> **First**, regarding the question about level-0 ciphertext, we do not require bootstrapping for level-0 ciphertext to continue the HE computation and just decrypt the ciphertext at the end of the client. As we mentioned in the conclusion, we use **Leveled Homomorphic Encryption (LHE), instead of Fully Homomorphic Encryption (FHE)**,  to build HE-based Privacy-Preserving Machine Learning. More accurately, it refers to the bootstrapping-free CKKS. We will further clarify this in the introduction.
>
> **Second**, we agree with the reviewer that bootstrapping has an enormous computational overhead, which could dominate the computation time if an excessive number of bootstrapping are needed for the network inference. In the meantime, as recognized by the reviewer in the comments, our proposed approach SPENCNN improves the computational efficiency by reducing the costly rotation operations for CKKS-based convolution, which is an approach orthogonal to the approaches that seek to reduce bootstrapping costs. (How our method can significantly contribute to the HE computations when bootstrapping is considered is thoroughly discussed in responding to reviewer A (the second point) as well).
>
> It is well-known that the HE computation cost is several orders of magnitudes higher than that in the plaintext space. Therefore, we believe that optimization efforts in different aspects must be pursued to make it really feasible in practice. Moreover, our proposed approach can be readily combined with the approaches (such as HEMET [1]) in reducing the bootstrapping computation overhead. For example, we can first employ HEMET to optimize the SqueezeNet [3] and save the redundant layer cost. Then, we can use SPENCNN to optimize the rest of the convolutional layers to further decrease the required rotations. We would like to add an additional section for discussion of this work with more details, and how these methods could be combined with our proposed approach to optimize the convolution in the revision of our manuscript.
>
> **Third**, we agree with the reviewer that using the fine-grained polynomial approximation could be an effective way to achieve higher model accuracy. As commented by the reviewer, this is an issue independent of our proposed techniques. In the meantime, a high-precision polynomial approximation for ReLu may also substantially increase the multiplicative levels unnecessarily. For example, methods proposed in [6] can also achieve ignorable accuracy loss by replacing the ReLUs with degree-2 polynomial approximations for deep neural networks (e.g. ResNet18, ResNet34). One high-precision approximation consumes as many as 14 levels [7] instead of 2 levels of degree-2 polynomial approximation, which increase the unnecessary bootstrapping budget for inference of a deep neural network.
>
> According to Figure 9 in [2], we can see that about 15 levels could be recovered by one-time bootstrapping. Using the settings in [2], one ciphertext needs 18 bootstrapping for ResNet20. Note that if a deep neural network is fine-tuned by the method in [6], the required number of bootstrapping for one ciphertext is 3 compared with 18 in [1]. Thus, the latency for bootstrapping will be in the same range with convolutional operations (see our estimated results in the following table. Thus, our SPENCNN’s contribution to Deep Neural Network with FHE (with bootstrapping) will achieve much more significant savings in HE computation.
>
> | APP_TYPE | ConvBN | BOOT | APPRELU |
> |---|---|---|---|
> | High-precision [7] | 346s | 1651s | 257s |
> | Degree-2 [6] | 346s | 275s | 16s |
>
> **Reference:**
>
> [1] HEMET: A Homomorphic-Encryption-Friendly Privacy-Preserving Mobile Neural Network Architecture, ICML 2021
>
> [2] Low-Complexity Deep Convolutional Neural Networks on Fully Homomorphic Encryption Using Multiplexed Parallel Convolutions, ICML 2022
>
> [3] Iandola, Forrest N., et al. "SqueezeNet: AlexNet-level accuracy with 50x fewer parameters and< 0.5 MB model size." arXiv preprint arXiv:1602.07360 (2016).
>
> [4] Kim, Miran, et al. "HEAR: human action recognition via neural networks on homomorphically encrypted data." arXiv preprint arXiv:2104.09164 (2021).
>
> [5] Lee, Joon-Woo, et al. "Privacy-preserving machine learning with fully homomorphic encryption for deep neural network." IEEE Access 10 (2022): 30039-30054.
>
> [6] Peng, Hongwu, et al. "PolyMPCNet: Towards ReLU-free Neural Architecture Search in Two-party Computation Based Private Inference." arXiv preprint arXiv:2209.09424 (2022).
>
> [7] Lee, Junghyun, et al. "Precise approximation of convolutional neural networks for homomorphically encrypted data." arXiv preprint arXiv:2105.10879 (2021).

---

### Official Review · Reviewer_CCNN · 2022-10-24

**Confidence:** 3
**Correctness:** 3
**Technical Novelty And Significance:** 3
**Empirical Novelty And Significance:** 4
**Recommendation:** 5

**Clarity, Quality, Novelty And Reproducibility:**

The technical contribution is described solely in figures and examples. Neither the appendix nor the supplemental material provide further insight. Similarly, I don't understand Algorithm 1 because it's so terse.

I'm missing a description of HEFNet.

Minor issues:
p2: missing whitespace after footnote 1
p3: "A client could upload his/her (...) data (...) for obtaining an online inference service." - Maybe "obtain the inference result"?
p3: "semi-honest (e.g. honest but curious)" - The two terms are synonyms and not an example.
p4: "pruning(Han", "pruning(Wen" - missing whitespace before brackets
p5: suggest G|M instead of "G%M=0"
p6: "algorithm 1" - Algorithm 1
p7: "state-of-the-art method. y We adopt"
p9: missing and misplaced whitespace in the caption of Table 4
p9: "There shall exist potential optimizations" - maybe "We expect potential optimizations"?


**Strength And Weaknesses:**

The core idea is interesting and novel to me, but I see issues with the presentation.


**Summary Of The Paper:**

The paper presents a method for privacy-preserving neural network inference using homomorphic encryption. The main contributation is optimizing the convoluational layer to fit the SIMD structure of homomorphic encryption, including reducing the number of non-zero weights and re-training.


**Summary Of The Review:**

Interesting idea marred by editorial issues

---

> ### Author Response · Authors · 2022-11-17
> **Response to Reviewer CCNN**
>
> We would like to thank reviewer CCNN for his/her constructive review comments on our work and favorable comments for the novelty of our research. For the description of HEFNet, it is included in Appendix A2. For Algorithm 1, we will provide the necessary descriptions in the maintext. we will fix all editorial issues as the reviewer mentioned in the comments. We sincerely hope the reviewer can kindly appreciate the technical contributions of our work.

---

### Official Review · Reviewer_AXmR · 2022-10-29

**Confidence:** 5
**Correctness:** 3
**Technical Novelty And Significance:** 3
**Empirical Novelty And Significance:** 3
**Recommendation:** 5

**Clarity, Quality, Novelty And Reproducibility:**

- The paper is clear for the most part. The figures are a bit challenging to understand, but becomes clear with the description in the text.

- The quality of the paper is good for the most part, however there are important baselines that are missing. Such methods have not been cited, discussed, or compared against.

- The proposed method is fairly novel. Most existing CNN implementations in FHE using standard dense convolutional layers. This paper proposes to use group convolutions which are more HE friendly. Sparsity pattern is also optimized for minimizing rotations as opposed to other criterion used in standard networks.

- The proposed approach is not reproducible based on the descriptions in the paper. There is missing information, hyper-parameters etc. And the paper does not provide code, nor do the authors promise to release code publicly later on.

**Strength And Weaknesses:**

Strengths:
- The paper rightly identifies the main bottleneck of *naive* implementations of convolution in FHE, namely rotations. As such, efforts to improve efficiency of convolution is necessary. Leveraging alternative convolutions, such as group convolutions or in the extreme depth-wise convolutions is interesting.
- Adopting pruning for sparsifying the convolution and optimizing the sparsity pattern for reducing number of homomorphic rotations.

Weaknesses:
- The main drawback of the paper is the lack of comparisons to prior work that improve efficiency of convolutions. These include multiplexed convolutions [1], mobile networks explored HEMET [2].
- The main premise of the paper is that rotations in convolutional layers are the main computational bottleneck of networks in FHE. So the paper considers shallow networks only, which are not likely to be practically useful. For instance 85% accuracy on CIFAR-10 is quite poor by the standards of the best plaintext models which achieve ~99% accuracy.
- As networks become deeper, the main accuracy bottleneck is low-degree polynomial approximations of non-linear functions like ReLU and the main computational bottleneck is the bootstrapping operations required for evaluating high-multiplicative depth circuits. So improving efficiency of convolutional layers does not benefit deeper networks since convolution is not the main bottleneck for such networks.

Other Clarification Questions:
- The paper does not mention how pooling operations or strided convolutions are handled. Strided convolutions result in wasted slots. How  does that affect the proposed convolutions?

[1] HEMET: A Homomorphic-Encryption-Friendly Privacy-Preserving Mobile Neural Network Architecture, ICML 2021
[2] Low-Complexity Deep Convolutional Neural Networks on Fully Homomorphic Encryption Using Multiplexed Parallel Convolutions, ICML 2022


**Summary Of The Paper:**

The paper seeks to improve the computational efficiency of convolution layers in FHE. Since homomorphic rotations are the primary computational bottleneck of convolutions in FHE, the paper seeks to reduce the number of rotations. This is achieved in two steps, 1) adopting group convolutions, which reduces out-level rotations, and 2) weight pruning, which reduces inner-level rotations.

The efficiency of the proposed convolution is evaluated on *shallow* CNNs designed for MNIST and CIFAR-10. The proposed approach shows appreciable speed-up over *naive* implementations of convolutions in FHE.

**Summary Of The Review:**

The paper proposed to use group convolutions and a weight pruning to mitigate the computational bottlenecks of convolutional layers, namely homomorphic rotations. The paper, however, does not compare to or discuss existing attempts toward HE-friendly CNNs. Furthermore, the experiments are conducted on shallow networks. The proposed approach will not provide much computational benefit for deeper networks since convolution is not the main bottleneck for such networks. Reproducibility is also limited.

Overall, the paper has good ideas, but an evaluation, comparison, and discussion of the broader utility of the proposed approach are missing.

**Update After Rebuttal:** The author's rebuttal does not adequately address the comments from the initial review. In theory, the method may have promise in achieving the claims in the rebuttal, but I do not believe it is straightforward and needs to be demonstrated. I will maintain the original rating.

---

> ### Author Response · Authors · 2022-11-17
> **Response to Reviewer AXmR**
>
> We would like to thank reviewer AXmR for the dedicated review comments on our work and for appreciating the novelty of our research. In what follows, we will present our response to the reviewer's comments and concerns.
>
> **First**, we would like to thank reviewer A to point us to the relevant work in reference [1] and [2]. Overall, our method is orthogonal to the proposed techniques in these works. **In [1], HEMET** improves the efficiency of CKKS-based convolution based on the saving multiplicative levels (**or rather reduce the layer numbers**) used in the total computation path, where our SPENCNN improves the computational efficiency by reducing the costly rotation operations for CKKS-based convolution. **These two approaches are orthogonal and can be combined together to improve HE-CNN latency**. For example, we can first employ HEMET to optimize the SqueezeNet and save the redundant layer cost. Then, we can use SPENCNN to optimize the rest of the convolutional layers to further decrease the required rotations. For the related work-**Multiplexed Parallel Convolutions[2]**, the multiplexed packing format used in [2] (inherited from HEAR [4]) is designed to “repair” the wasted slots of ciphertext after avg pooling and strided convolution. This method would save the number of ciphertexts, which correspondingly reduces the number of rotations used in convolution. Again, SPENCNN is orthogonal to the approach, and could integrate with these multiplexed packed ciphertexts. Consider an example, after an average pooling layer with stride=2, we have 4 group-interleaved encoded ciphertexts which all have wasted neighbor slots. We can pack the 4 different ciphertexts into 1 ciphertext by the multiplexed packing pre-processing step (based on HEAR [4)]. Then we can still perform the same HE-group convolution. After that, the post-processing step [4] to aggregate these neighbor channels can be conducted. Hence our group-interleaved convolution technique basically re-uses these standard procedures without requiring any special changes. For the proposed sub-block pruning step, we just need to identify the sub-blocks based channels on the same ciphertext after multiplexed packing. We would like to add an additional section for discussion of these works with more details, and how these methods are orthogonal to ours, and can further be integrated with our proposed approach to optimize the convolution in the new revision of our manuscript.
>
> **Second**, we agree with review A that bootstrapping incurs an enormous computational overhead, which could dominate the computation time if an excessive number of bootstrappings are needed for the network inference. However, we cautiously disagree with reviewer A that a high-precision polynomial approximation [7] for ReLu --- one of the primary reasons that increases the multiplicative levels — is always necessary to ensure the model accuracy. Note that, the high-precision polynomial approximation for one level of ReLU in [2] and [5] would consume as many as **14 multiplicative levels**. This introduces tremendous level consumption which also leads to significantly more bootstrappings required. In contrast, other methods (such as the one in [6]) that replace all ReLUs by degree-2 polynomial function for deep neural networks (e.g. ResNet34, ResNet18) **with ignorable accuracy loss compared with pre-train models, and a degree-2 polynomial approximation only introduces two multiplicative levels**. According to Figure 9 in [5], we can see that about 15 levels could be recovered by one-time bootstrapping. Using the settings in [5], one ciphertext would need 18 bootstrappings for ResNet20. If we use degree-2 polynomial approximation instead of the high-precision approximation adopted by [2,5,7], one ciphertext only requires **3 times (instead of 18 times)** bootstrapping, **which puts the latency for bootstrapping in the same range with the latency for convolution based on our profiling results in the same hardware environment as that of [7]** (ResNet-20. Single thread, same CPU), as shown in the following Table 1. Note the latency of RELU is now based on degree 2 polynomials. While this work targets Leveled-HE (LHE)-CKKs without involving bootstrapping, these results indicate that our solution is practically meaningful regardless of with or without bootstrapping. The achievable efficiency can be further boosted for deeper networks due to involving more convolutions. Furthermore, as Reviewer WT91 pointed out, optimizing the costly HE rotation of convolution operations and bootstrapping are two independent directions. They are both important based on our above results and analysis. We will incorporate this discussion and cite the related works in the revision based on the reviewers’ advice.

---

> > ### Author Response · Authors · 2022-11-17
> > **Additional Response to Reviewer AXmR**
> >
> > **Table 1**
> > | APP_TYPE | ConvBN | BOOT | APPRELU |
> > |---|---|---|---|
> > | High-precision [7] | 346s | 1651s | 257s |
> > | Degree-2 [6] | 346s | 275s | 16s |
> >
> > **Third**, for our handling of the pooling operations and strided convolutions, we do not repair the “wasted slots” as that of [2] (or HEAR [4]). Instead, we leave the wasted slots as is. Because these slots would not affect our HE-group convolution in implementation by simply multiplying zero with them in the corresponding slots of weight plaintext in the next convolution layer.
> >
> >
> > **Four**, to address Review A’s concern on reproducing the experimental results, we would like to include the hyperparameter of training and retraining the model in the appendix in the later revision and **also promise to release the code publicly to make our contributions to the whole research community.**
> >
> > **Reference:**
> >
> > [1] HEMET: A Homomorphic-Encryption-Friendly Privacy-Preserving Mobile Neural Network Architecture, ICML 2021
> >
> > [2] Low-Complexity Deep Convolutional Neural Networks on Fully Homomorphic Encryption Using Multiplexed Parallel Convolutions, ICML 2022
> >
> > [3] Iandola, Forrest N., et al. "SqueezeNet: AlexNet-level accuracy with 50x fewer parameters and< 0.5 MB model size." arXiv preprint arXiv:1602.07360 (2016).
> >
> > [4] Kim, Miran, et al. "HEAR: human action recognition via neural networks on homomorphically encrypted data." arXiv preprint arXiv:2104.09164 (2021).
> >
> > [5] Lee, Joon-Woo, et al. "Privacy-preserving machine learning with fully homomorphic encryption for deep neural network." IEEE Access 10 (2022): 30039-30054.
> >
> > [6] Peng, Hongwu, et al. "PolyMPCNet: Towards ReLU-free Neural Architecture Search in Two-party Computation Based Private Inference." arXiv preprint arXiv:2209.09424 (2022).
> >
> > [7] Lee, Junghyun, et al. "Precise approximation of convolutional neural networks for homomorphically encrypted data." arXiv preprint arXiv:2105.10879 (2021).

---

> > ### Comment · Reviewer_AXmR · 2022-11-18
> > **Quick Response**
> >
> > Thanks for responding to the review comments.
> >
> > I agree with the reviewers that at a conceptual level, the proposed group convolution+weight pruning is orthogonal to HEMET [1] and Multiplexed Parallel Convolutions [2] and potentially be combined with the network design in [6].
> >
> > However, I disagree that it is straightforward to practically realize it. *First*, [1] and [2] use dense convolutions, so simply replacing them with group convolutions and weight pruning may not preserve the accuracy of the model. *Second*, implementing SpENCNN within [1], [2] or [6] requires careful design of the homomorphic evaluation architecture. For example, the encryption parameters (n, q) need to be chosen carefully to ensure at least 128 bits of security, the bootstrapping precision needs to be sufficiently large enough to prevent attacks, placement of bootstrapping must be careful to prevent wasted levels and realize practical speedups etc.
> >
> > Embedding SpENCNN into these existing solutions [1], [2], or [6], and demonstrating the real utility of SpENCNN would go a long way towards understanding the impact of SpENCNN on accuracy and latency. But, without an actual practical demonstration, it is difficult to judge whether the claims in the rebuttal can be realized.

---

> > > ### Author Response · Authors · 2022-11-18
> > > **Response to Reviewer AXmR**
> > >
> > > We thank you for your quick response and agree with us that our technique is orthogonal to existing solutions in [1][2][6]. We also appreciate your detailed comments and thoughts on the issues of integrating our technique with these existing orthogonal solutions.
> > >
> > > However, we cautiously disagree with the reviewer that our SpENCNN is questionable because of not practically implementing and demonstrating its effectiveness by embedding it into these existing orthogonal solutions. Because 1) further designing better solutions by combining ours with many existing solutions is not the focus of the work, and it is beyond the scope of this work; Reference [6]-Multiplexed Parallel Convolution with bootstrapping (ICML22 published in July 2022) can be considered as a contemporaneous paper according to ICLR rule. 2) Our rebuttal clearly shows the feasibility and benefit of integration (including the promising results like Table 1 to show its effectiveness under bootstrapping).
> > >
> > > First, in our view, it is common and reasonable that some efforts are needed to put together orthogonal techniques for better performance. For example, parameters like group number and pruning ratio should be tuned to balance the accuracy and HE operation number reduction when applying them to dense convolutions, we showed similar trade-off results in Table 1 of the evaluation section in the paper. Also, like other works, different choices of security parameters, bootstrapping precision, and location will impact the effectiveness of our solution. . However, this should not impact (or dilute) the novelty and contributions of this work itself in principle (as our initial promising results show in the rebuttal). Indeed, we believe how to practically integrate our work with the (many) existing orthogonal solutions inspires interesting research problems for the future (just as the example questions mentioned by the reviewer and is beyond the scope of this work.
> > >
> > > Second, for reviewers’ mentioned work [6]-Multiplexed Parallel Convolution with bootstrapping, it is officially published in ICML2022 in July 2022. According to ICLR’s rule, it can be considered as a contemporaneous paper (published on or after May 28, 2022). While we very much thank the reviewer to point us to this important work and will cite and discuss the effectiveness as much as we can. Based on our understanding of the rule, authors maybe excused for not knowing the work. https://iclr.cc/Conferences/2023/ReviewerGuide (Q: Are authors expected to cite and compare with very recent work? What about non peer-reviewed (e.g., ArXiv) papers? (updated on 7 November 2022)).
> > >
> > > We sincerely hope the reviewer can consider these factors when making the final decision. Thank you again!

---

### Decision · Program_Chairs · 2023-01-20

**Decision:**

Reject

**Justification For Why Not Higher Score:**

Not good enough.

**Justification For Why Not Lower Score:**

NA

**Metareview: Summary, Strengths And Weaknesses:**

The paper presents way to make computational speedups for FHE in convolutional layers. There are specific aspects of FHE that are extremely slow and the paper makes progress towards reducing costs by some pruning and group convolutions.

Some experiments are presented with shallow CNNs for MNIST and CIFAR. However in terms of accuracy the drop is significant. More comparison prior work and more empirical evaluations are needed. There are some interesting observations in the responses to the reviews, but in my opinion they need to be incorporated properly in the paper and  the paper resubmitted at a suitable venue.